# Profiles of Volatile and Phenolic Compounds as Markers of Ripening Stage in Candonga Strawberries

**DOI:** 10.3390/foods10123102

**Published:** 2021-12-14

**Authors:** Rosaria Cozzolino, Bernardo Pace, Michela Palumbo, Carmine Laurino, Gianluca Picariello, Francesco Siano, Beatrice De Giulio, Sergio Pelosi, Maria Cefola

**Affiliations:** 1Institute of Food Science, National Research Council (CNR), Via Roma 64, 83100 Avellino, Italy; carmine.laurino@isa.cnr.it (C.L.); gianluca.picariello@isa.cnr.it (G.P.); francesco.siano@isa.cnr.it (F.S.); beatrice.degiulio@isa.cnr.it (B.D.G.); 2Institute of Sciences of Food Production, National Research Council of Italy (CNR), c/o CS-DAT, Via M. Protano, 71121 Foggia, Italy; michela.palumbo@ispa.cnr.it (M.P.); sergio.pelosi@ispa.cnr.it (S.P.); maria.cefola@ispa.cnr.it (M.C.); 3Department of Agriculture, Food, Natural Resources and Engineering, University of Foggia, Via Napoli 25, 71122 Foggia, Italy

**Keywords:** *Fragaria × ananassa* Duch., Sabrosa, ripening stage, headspace solid phase microextraction (HS SPME GC/MS), HPLC-MS/MS principal component analysis

## Abstract

Volatile compounds, quality traits (total phenols and antioxidant capacity) and High-performance liquid chromatography (HPLC)-isolated polyphenols of strawberries, variety Sabrosa, commercially referred to as “Candonga”, harvested at three different times (H1, H2 and H3) and at two different ripening stages, namely half-red (Half-red-H1, Half-red-H2 and Half-red-H3) and red (Red-H1, Red-H2 and Red-H3) were evaluated. Dominant anthocyanins, namely cyanidin-3-*O*-glucoside, pelargonidin-3-*O*-glucoside and pelargonidin-3-*O*-rutinoside, as well as *p*-coumaryl hexoside increased during harvesting, differently from flavonoids, such as quercetin-3-*O*-glucoside, kaempferol-3-*O*-glucoronide and quercetin 3-*O*-glucoronide, that declined. Samples clustered in different quadrants of the principal component analysis (PCA) performed on volatiles, quality traits and phenolic compounds, highlighting that only the red samples were directly correlated to volatile components, as volatiles clearly increased both in number and amount during ripening. In particular, volatiles with a positive impact on the consumers’ acceptance, including butyl butyrate, ethyl hexanoate, hexyl acetate, nonanal, terpenes and lactones, were positively associated with the Red-H1 and Red-H2 strawberries, while volatiles with negative coefficients related to consumer liking, including isopropyl butyrate, isoamyl butyrate and mesifurane directly correlated with the Red-H3 samples. Accordingly, strawberries harvested at Red-H1 and Red-H2 ripening stages could be preferred by the consumers compared to the Red-H3 fruit. Altogether, these results could help to individuate quality traits as putative markers of the ripening stage, and optimize the process of post-harvesting ripening to preserve or improve the desirable aromatic characteristics of strawberries.

## 1. Introduction

Strawberry (*Fragaria* × *ananassa* Duch.) belongs to the Rosaceae family, and is one of the most commonly consumed berry fruit and cash crops worldwide, with more than 2000 varieties [1]. Because of the high content of ellagic acid (EA) and its precursors, strawberries are considered functional foods [1]. 

The functional traits of the strawberry originate from to the combination of vitamin C and other antioxidant components, primarily flavonoids, anthocyanins and EA. In spite of their relative low content (0.001–0.01% of fresh fruit weight), the volatile organic compounds (VOCs) are the main responsible for the strawberries’ flavor, which is a crucial factor in determining the consumer’s preference and the sensory quality of the fruit [2,3,4]. Strawberry fruit can be considered a typical example of a complex fruit aroma, since several hundred VOCs concur to determine the flavor of fresh strawberries [4]. 

Qualitative and quantitative profiles of strawberry VOCs stem partially from genetic traits and growth-dependent activation of specific metabolic pathways, so that they show specific patterns and distinctive volatile components depending on the cultivar and degree of ripening [5]. Thus, aroma can be a fingerprint to distinguish among varieties and stage of fruit development [3].

The content of VOCs increases during the maturation in climacteric fruits, and a similar effect could be expected in non-climacteric fruit as well, such as the strawberry [3].

Harvesting fruit before it is fully ripe is a common practice for many fruits in the supply chain, as it allows to complete ripening in post-harvest storage [6].

Clearly, the maturity stage affects the consumer liking of these fruits, including strawberries, as a consequence of the changes affecting the volatile profile [7]. 

Monitoring the VOCs pattern during the maturation process could offer specific markers for establishing the optimal harvest time, ensuring a standardized aroma to consumers as well as maximizing quality and phytosanitary characteristics, thus contributing to minimizing post-harvest losses [5]. Volatile esters are the compounds most associated with strawberry fruit ripening, although VOCs belonging to different classes could be considered markers of maturity, varying with the cultivar [3]. 

Among the phenolic compounds, anthocyanins are responsible for the bright red color of the strawberries. They are the most abundant phenolic compounds in most of the 27 cultivars analyzed by Aaby et al. [8], and are known to possess health benefits. The anthocyanins also vary with cultivars, and steadily increase during ripening, as assessed by analyzing the number of strawberry cultivars [8]. 

The present study aimed at evaluating for the first time the changes in fruit quality traits, content of VOCs and phenolic compounds in strawberry samples of the variety Sabrosa, commercially referred to as “Candonga”. Strawberries were collected at two different ripening stages (half-red and red) in three different harvesting times. The goal was to gain information about the modification at a metabolite level occurring during strawberry maturation and the contributing individuate putative molecular markers, in order to optimize the selection of the timing of harvest. 

## 2. Materials and Methods

### 2.1. Plant Material

“Candonga” strawberries (*Fragaria* × *ananassa* Duch. var. Sabrosa) were harvested by Apofruit (Scanzano Jonico, Italy) on 21 May (first harvest time, H1), 27 May (second harvest time, H2) and 1 June (third harvest time, H3) at two different ripening stages, namely half-red (in ripening phase, fully expanded and 50% red, indicated as Half-red-H1, Half-red-H2 and Half-red-H3) and red (in ripening phase, fully expanded and 100% red, indicated as Red-H1, Red-H2 and Red-H3), according to visual criteria (Figure 1).

Half-red and red strawberries showed soluble solids at harvest of about 8.8 ± 0.7 and 9.8 ± 0.2 °Brix, respectively. At each harvest time, fruit berries were packed into PET trays (Cartonpack spa, Rutigliano, Ba) (about 500 g for each tray) and transported in refrigerated conditions (4 ± 1 °C) to the Consiglio Nazionale delle Ricerche (CNR) laboratories, and were analyzed within 3 h since harvesting. Here, strawberries were visually inspected and selected to choose fruits free of physical or biological damage. Respiration rate titratable acidity, total soluble solids, pH, color parameters, antioxidant activity, total phenols, HPLC-separated individual phenolic compounds and VOCs were evaluated at each harvest time and for each ripening stage. 

### 2.2. Reagents and Chemicals

Chemicals, standards and reagents were from Sigma-Aldrich (St. Louis, MO, USA). Folin–Ciocalteu’s phenol reagent was purchased from Merck (Darmstadt, Germany). Ultra-pure water (resistivity at 25 °C of 18 MΩ cm) was from a Millipore Milli-Q purification system (Millipore Corp., Bedford, MA, USA), while helium at a purity of 99.999% (Rivoira, Milan, Italy) was used as GC carrier gas. The HS-SPME fibers and the glass vials were purchased from Supelco (Bellofonte, PA, USA); the capillary GC-MS column High Polarity (HP)-Innowax (30 m × 0.25 mm × 0.5 μm) was acquired from Agilent J&W (Agilent Technologies Inc., Santa Clara, CA, USA).

### 2.3. Respiration Rate

The respiration rate of strawberries was measured at 8 °C using a closed system, according to the method reported by Kader [9]. In particular, for each ripening stage and replicate (*n* = 3), about 500 g of sample were put into 3.6 L sealed plastic jar (one jar for each replicate) where CO_2_ was allowed to accumulate up to 0.1% of the standard concentration of the CO_2_. The time taken to get to this value was detected by measuring the CO_2_ amounts at regular intervals of time. The CO_2_ analysis was conducted by injecting 1 mL of gas sample from the headspace of the plastic jars through a rubber septum into a gas chromatograph (p200 micro GC-Agilent, Santa Clara, CA, USA) fitted with dual columns and a thermal conductivity detector. CO_2_ was analyzed with a retention time of 16 s and a total run time of 120 s using a 10 m porous polymer (PPU) column (Agilent, Santa Clara, CA, USA) at a constant temperature of 70 °C. Respiration rate was reported as mL CO_2_/kg h.

### 2.4. Total Soluble Solids, Titratable Acidity and pH 

For each replicate and at each ripening stage, about 100 g of strawberries was homogenized to obtain the fruit juice which was used to assay total soluble solids (TSS), titratable acidity (TA) and pH. The TSS content was determined using a digital refractometer (DBR35-XS Instruments, Carpi, Italy) and results were expressed in °Brix. 

The pH of the fruit juice was determined using a pH meter (PH-Burette 24-Crison Instrument, Barcelona, Spain) and the titratable acidity (% citric acid) was measured by titration using 0.1 M NaOH to the final pH 8.1, revealed with phenolphthalein as the indicator.

### 2.5. Antioxidant Activity and Total Phenolic Content

The analyses of the antioxidant activity (AA) and the total phenolic content (TPC) of strawberry samples were carried out on samples extracted as follows: for each replicate and at each ripening stage, 5 g of strawberries (chopped into small pieces) was homogenized in 20 mL methanol/water solution (80:20 *v*/*v*) for 2 min, using a homogenizer (T-25 digital ULTRA-TURRAX^®^-IKA, Staufen, Germany) and then centrifuged (Prism C2500-R, Labnet, Edison, NJ, USA) at 15,000 rpm for 5 min at 4 °C. The extracts were collected and stored at −20 °C before the analysis.

The AA was measured on the methanol extract using the DPPH (1,1-diphenyl-2-picrylhydrazyl) assay as described by Cefola et al. [10]. The absorbance was measured at 515 nm after 40 min in the dark, using a spectrophotometer (UV-1800, Shimadzu, Kyoto, Japan). The results were expressed as mg of Trolox per 100 g of fw (fresh weight) using a Trolox calibration curve (82–625 μM; R^2^ = 0.999).

The TPC was determined according to Fadda et al. [11]. In detail, 100 μL of each extract was mixed into 1.58 mL of water, 100 μL of Folin–Ciocalteu’s reagent and 300 μL of sodium carbonate solution (200 g/L). The resulting absorbance was measured at 765 nm after 2 h in the dark and the results were expressed as mg of gallic acid equivalents (GAE) per 100 g of fw. The calibration curve of gallic acid was prepared with five points, from 50 to 500 μg/mL, with R^2^ = 0.998.

### 2.6. Analysis of Polyphenols Compounds

#### 2.6.1. Reversed Phase-High Performance Liquid Chromatographic-Diode Array Detector (RP-HPLC-DAD) Semi-Quantitative Determination of Polyphenols 

Strawberry methanolic (80%, *v*/*v*) extracts, prepared as described above (Par. 2.5), were ten-fold diluted with aqueous 0.1% (*v*/*v*) trifluoroacetic acid (TFA) and 100 µL of the diluted sample was separated using a modular HP 1100 chromatographer (Agilent Technologies, Paolo Alto, CA, USA) equipped with a 250 × 2.0 mm i.d. C18 reversed-phase column, 4 mm particle diameter (Jupiter Phenomenex, Torrance, CA, USA) held at 37 °C in a thermostatic oven. HPLC runs were performed at a constant flow rate of 0.2 mL/min applying the following gradient of solvent B: isocratic elution at 5% B for 5 min, 5–60% linear gradient of B for 5–65 min and 60–100% B at 65–70 min. Eluent A and B were 0.1% TFA in HPLC-grade water and 0.1% TFA in acetonitrile, respectively. Samples were run in triplicate and monitored at wavelengths λ = 520, 360, 320 and 280 nm using a diode array detector (DAD), also acquiring a UV-Vis spectrum every second in the 200–700 nm range. A home-prepared multicomponent standard solution containing 15 (poly)phenols among which were gallic acid, *p*-coumaric acid, quercetin 3-*O*-glucoside, rutin, EA and quercetin and kaempferol aglycones (all from Sigma-Aldrich, St. Louis, MI, USA), was used to confirm or exclude the assignment of some phenolic compounds. Flavonoids were semi-quantified by plotting the area of peaks integrated at 360 nm on an external calibration curve built with standard rutin at a known concentration in the 0.10–5.00 μg/mL range (R^2^ = 0.99). Antocyanins and *p*-coumaryl-hexoside were semi-quantified based on calibration curves built with cyanidin-3-*O*-glucoside and *p*-coumaric acid with absorbance at 520 and 320 nm, respectively. Data were processed using the ChemStation software (version A.10) purchased with the chromatograph. Analyses were carried out in triplicate and peak area values were averaged.

#### 2.6.2. Nanoflow HPLC-ESI MS/MS Analysis

Identification of the phenolic compounds were confirmed by nanoflow-HPLC ESI MS/MS analysis, which was performed using an Ultimate 3000 ultra-high performance liquid chromatography instrument (Dionex/Thermo Scientific, San Jose, CA, USA), online coupled with a Q Exactive Orbitrap (Thermo Scientific) mass spectrometer, using previously detailed conditions [12]. The mass spectrometer switched between positive and negative ionization polarity in 1 s and scanned the 120–1200 *m*/*z* range, and operated in data-dependent acquisition for MS/MS, fragmenting up to 5 most intense signals in 1 s with 10 s of dynamic exclusion. Spectra were elaborated using the Xcalibur Software 3.1 version (Thermo Scientific).

### 2.7. Volatile Organic Compounds (VOCs) Analysis 

#### 2.7.1. Sample Preparation and HS SPME Procedure

The HS SPME conditions of analysis were optimized assaying strawberry samples obtained from a local supermarket. Profiling of VOCs was performed by HS SPME/GC-MS according to Zorrilla-Fontanesi et al. [13], but utilizing a DVB/CAR/PDMS (50/30 mm) fiber, with 50 °C as the extraction temperature and 20 min as the extraction time. Concerning the sample preparation, 1 g of “Candonga” strawberry sample was put into a 20 mL screw-on cap HS vial and mixed into 0.3 g of NaCl. To ensure the analytical reproducibility, 1.5 μL each sample was spiked with 20 ppm of 2-octanol taken from a stock solution, used as the internal standard (IS). Vials were then sealed with a Teflon septum and an aluminum cap (Chromacol, Hertfordshire, UK) and stirred. The equilibration time and temperature were 10 min and 40 °C, respectively. The extraction and injection phases were automatically performed using an autosampler MPS 2 (Gerstel, Mülheim, Germany). Afterwards, the HS-SPME fiber was automatically introduced into the vial’s septum for 20 min to allow VOCs to be adsorbed onto the fiber surface.

#### 2.7.2. Gas Chromatography-Quadrupole Mass Spectrometry Analysis (GC-qMS)

VOC analysis was performed using a gas chromatograph model GC 7890A coupled to a mass spectrometer 5975 C (system from Agilent Technologies, CA, USA). The HS SPME fiber was inserted for 10 min into the injector port of the GC instrument. VOCs were thermally desorbed and directly transferred to a capillary column HP-Innowax. Oven temperature conditions were initially set at 50 °C for 3 min, then increased to 160 °C at 5 °C min^−1^, held at 160 °C for 1 min, ramped to 250 °C at 10 °C min^−1^ and stable at 250 °C for 2 min. VOCs were analyzed at an ionization energy of 70 eV and detected by mass selective detector. The detector operated in a mass range between 30 and 300 u with a scanning speed of 2.7 scans/s. VOCs were identified by mass spectra through matching with the standard NIST05/Wiley07 libraries, by comparing the retention indices (RI) (as Kovats indices) with literature data and pure standards when available. Each sample was analyzed in triplicate with a randomized sequence in which blanks were also recorded. For each volatile component, the peak area was calculated from the total ion chromatogram (TIC) and semi-quantified by relative comparison with the peak area of the IS (Relative Peak Area, RPA%).

### 2.8. Statistical Data Analysis

For each harvest, the effect of the ripening stage (half-red or red) on respiration rate, TA, TSS, pH, AA, TPC and VOCs was evaluated by performing a one-way Analysis of variance (ANOVA)for *p* ≤ 0.05. The mean values (*n* = 3) were separated using the least significant difference (LSD) test (*p* ≤ 0.05), and Statgraphics Centurion (version 18.1.12, Warrenton, VA, USA) was used for statistical analyses.

To highlight the VOCs correlated to half-red and red strawberries, a principal component analysis (PCA) was carried out using the software Statistica version 6.0. (Statsoft Inc., Tulsa, OK, USA). 

## 3. Results and Discussion

### 3.1. Quality Traits in Half-Red and Red Strawberries

Respiration rate did not significantly differ between the two ripening stages, presenting values of about 15.10 ± 1.3 mL CO_2_ kg^−1^ h^−1^, in line with those observed in the classification reported by Kader [14] (Table 1). As concerns the physical parameters, the one-way ANOVA showed statistically significant different values (*p* ≤ 0.0001) between the ripening stages at H1, H2 and H3. In detail, TA in red strawberries was lower than that detected in half-red samples in all harvest times. These results were consistent with the lower pH values detected in half-red strawberries compared to those of the red samples at all the harvest times (Table 1). The decrease in TA with the consequent increase in pH during the ripening has been previously explained by the conversion of organic acids into sugars in the course of the respiration process [15]. The ripening stage significantly influenced TSS at each harvest time, showing higher values (13.8, 14.5 and 4.4% more in H1, H2 and H3, respectively) in red samples than in half-red ones (Table 1). Similar results of TA, pH and TSS have been already reported by Correia et al. [16] and Aguero et al. [17] in “Candonga” strawberries harvested in spring at full ripening.

The ripening stage of the strawberries significantly influenced AA, as half-red samples presented higher values (345.06 ± 4.2 mg Trolox 100 g^−1^ fw) than red berries (287.46 ± 14.2 mg Trolox 100 g^−1^ fw), considering the mean of the three harvest times (Table 1). These findings are in line with previous determination of DPPH radical scavenging capacity [18,19]. 

Analysis of TPC conducted on the strawberry samples at all the three harvest times mirrored the trend of AA, confirming that the AA values are directly related to the TPC, as already recorded in several fruit and vegetable crops [20,21,22]. Specifically, the half-red fruits displayed a TPC of 13.9% higher than the one detected on the red samples (Table 1), according to previous reports [18,19]. 

### 3.2. Phenolic Compounds in Half-Red and Red Strawberries

The comprehensive pattern of polyphenols in the strawberry appears very complex, as compounds belonging to several classes, including hydroxycinnamic acid derivatives, flavonoids, and anthocyanins are variously represented. Despite numerous studies carried out to characterize strawberry metabolites, the current inventory of phenolic compounds emerging from the literature is controversial because of the large number of variety of cultivars available and the diversity of analytical methods employed. Nevertheless, the fraction is substantially dominated by a few compounds, whereas a multitude of other metabolites occur at a minor abundance [23,24]. The RP-HPLC separation of phenolic compounds in strawberry methanol extracts has been monitored at multiple wavelengths. Typical RP-HPLC chromatograms of extracts from red (left A) and half-red (right B) “Candonga” strawberries are shown in Figure 2. In particular, in Figure 2, the HPLC chromatograms recorded at 280 nm for the general detection of phenolic compounds, at 360 nm for the selective detection of flavonoids and EA and at 520 nm for the diagnostic detection of anthocyanins are compared. The main HPLC peaks were assigned in Table 2, based on the converging indications coming from previous identification of strawberry phenolics [23,24,25] UV-Vis spectra acquired with the DAD, high-resolution MS and MS/MS spectra. In agreement with previous data, *p*-coumaryl-hexoside (peak 1, P1) was the most abundant hydroxycinnamic acid derivative of strawberry, better detected at 320 nm (not shown) because of the characteristic absorbance band centered at 315 nm. In half-red strawberries, P1 was 3-/4-fold lower than in the red samples. Similar to other strawberry cultivars, the anthocyanin profile of the strawberry is substantially conserved among the cultivars [25]. Pelargonidin-3-*O*-rutinoside (P4) was the prevalent anthocyanin both in red and half-red strawberry samples, followed by less abundant pelargonidin-3-*O*-glucoside (P3) and cyanidin-3-*O*-glucoside (P2). However, the concentration of anthocyanins in unripe fruits was nearly half than the red counterpart. The chromatogram at 360 nm was dominated by quercetin-3-*O*-glucuronide (P7), which was more abundant in Half-red-H1 and Half-red-H2 than in the ripened counterpart. Strawberries have been generally described as fruits rich in EA; however, the content of free EA can vary in a wide range, depending on the cultivar as well as on a series of abiotic factors [26]. In the current strawberry samples, free EA was detected at concentrations lower than the limit of quantification, co-eluting with kaempferol-3-*O*-glucoside (peak 6, P6), as also confirmed with a separate injection of pure EA. MS and MS/MS analysis allowed to establish the presence of free EA and assess that it was slightly more intense in half-red than in ripe strawberry [25] (data not shown). No glycosylated derivatives of EA were detected among the main compounds by RP-HPLC, while EA conjugates (i.e., pentoside and hexoside) were detected both in half-red and red strawberry by ion extraction in the MS runs (data not shown). Several minor signals of relatively high molecular weight compounds detected by LC-MS were likely ellagitannins that could be converted into EA by processing or chemical hydrolysis [25]. A detailed characterization of strawberry ellagitannins is challenging and requires dedicated investigations [24,27].

The semi-quantitative figures determined for the individual phenolic compounds were in line with most of the previous determinations reported by other authors [8].

### 3.3. VOCs Compounds in Half-Red and Red Strawberries

#### 3.3.1. Comparative Determination of VOCs in the “Candonga” Strawberry Samples at Two Different Ripening Stages

Overall, fifty-seven volatile compounds were identified by HS-SPME GC-MS analysis in the “Candonga” strawberry samples at two different ripening stages (half-red and red), and at three different harvest times which consisted of esters (26), aldehydes (5), alcohols (5), acids (9), terpenes (5), furanones (3), lactones (3) and others (1). 

One-way ANOVA performed on the semi-quantitative data (RPA%) evidenced significant qualitative and quantitative changes in the profile of VOCs over the course of the two maturation stages, as reported in Table 3. On the other hand, Appendix A includes the abbreviation code, the experimental and literature Kovats indexes and the identification methods for the assigned VOCs.

According to previous studies, ester compounds, responsible for the strawberry fruity and floral aroma, were the most abundant chemical class, with 18 and 26 different components accounting for about 37% and 49% of all the volatile compounds detected in the half-red and red fruit, respectively [3]. The majority of these esters have been previously reported in “Candonga” strawberries [28,29]. The most representative esters in half-red strawberries were methyl butyrate (E2) (37%), methyl hexanoate (E9) (27%) and *trans*-2-hexen-1-ol acetate (E16) (20%), which increased up to about 40% of the total amount of esters in the red fruit. At ripening, together with γ-decalactone (L2), E16 became the component with the highest concentrations, individually constituting nearly 20% of the total content of the VOCs. These findings suggest that E16 and L2 could be considered as key indicators of full maturity in this variety (Table 3). 

Butyl acetate (E6), methyl pentanoate (E7), ethyl pentanoate (E8), butyl butyrate (E10), ethyl hexanoate (E11), isoamyl butyrate (E12), octyl isobutyrate (E23) and octyl 2-methylbutyrate (E24) were detected only in red berries (Table 3). In fruit, enzymatic biosynthesis of volatile esters through the esterification of alcohols and acyl-CoA, derived from both fatty acid and amino acid metabolism, occurs during the late ripening steps [30]. The enzyme responsible for the final step of ester formation is alcohol acyltransferase (AAT), which in strawberries can exhibit a 16-fold increase from the half-red to the full red ripeness degree [31]. A consistent correlation occurs among the expression of an AAT gene, AAT activity and concentrations of some volatile esters [31]. The availability of substrates for the biosynthesis of volatile esters is believed to be a limiting step in the production of esters, leading to different flavor profiles at each ripeness stage [30,31,32]. 

Five aldehyde compounds were common to the half-red and red stages, although occurring at a significantly higher concentration in full red berries (Table 3). The formation of C6 aldehydes originates from the lipid oxidation pathway, which involves lipoxygenase (LOX) and hydroperoxide lyase (HPL) enzymes. During ripening, the content of the two C6 aldehydes, namely hexanal (Ald1) and 2-hexenal (Ald 2), increases, paralleling the 25–200% increment in the activity of LOX and HPL enzymes and the availability of total linolenic acid as the precursor, which rises up to 200% [32]. 

Nonanal (Ald3) and decanal (Ald5), arising from the autoxidation of oleic acid, and benzaldehydes (Ald4), deriving from Strecker degradation of aromatic amino acids, increase during oxidative processes [33]. 

Gene expression studies reported the existence of a specific enzyme, O-methyltransferase, that is responsible for the synthesis of mesifurane (F1). Additionally, a *Fragaria × ananassa* quinine oxidoreductase (FaQR) was documented to be involved in the formation of furaneol (F2). F1 and F2 increased considerably during ripening of strawberries, as the enzymes belonging to their biosynthetic pathway showed the maximum activity at the full red stage, allowing potential targets to engineer the strawberry flavor from breeding populations [31].

#### 3.3.2. Selection of the VOCs Correlated to Ripening Stage of Strawberry by Principal Component Analysis

Principal component analysis (PCA) took into account AA, TPC, the semi-quantitative data of both individual phenolic compounds (Table 2) and VOCs (Table 3), to infer possible significant associations of strawberries at the two different ripening stages (half-red or red) and at the three harvesting times (Figure 3).

The two components accounted for 94.8% of the variation in the dataset, since PC1 and PC2 explained 90.3% and 4.5% of the total variance, respectively. In the PCA plot, all red samples (Red-H1, Red-H2 and Red-H3) appeared in the left part of the score plot, while all the half-red strawberries (Half-red-H1, Half-red-H2 and Half-red-H3) were positioned on the right part of the graph, revealing a different distribution of the samples in the PCA quadrants (Figure 3A). Specifically, Half-red-H1 and Half-red-H2 samples clustered closely in the bottom right part of the plot, having positive PC1 and negative PC2 values (Figure 3A). These samples showed a significant correlation with three phenolic compounds (P5, P6 and P7) and with AC (Figure 3B). The direct association of Half-red-H1 and Half-red-H2 samples with the P5, P6 and P7 metabolites likely reflects the decline in the synthesis of flavonoids during the last stage of ripening, as demonstrated for other fruits. The decrease in flavonoids might correspond to their utilization for the downstream biosynthesis of other metabolites, or to the covalent association with other cellular components [34]. Furthermore, a higher contribution of flavonoids compared to other classes of phenolic compounds could explain the correlation with AA. 

On the contrary, Half-red-H3 fruit presented positive PC1 and PC2 scores (Figure 3A) and appeared statistically associated only with TPC (Figure 3B).

The strawberry samples classified as Red-H1 and Red-H2, with negative PC1 and PC2 scores, were grouped closely in the bottom left part of the PCA plot (Figure 3A) and covaried with the same volatile metabolites: 39 VOCs comprising 14 esters (E9, E10, E13, E15, E16, E17, E18, E19 E20, E21, E23, E24, E25, E26), 4 aldehydes (Ald2, Ald3, Ald4, Ald5), 3 alcohols (Al1, Al2, Al4), 8 acids (Ac1, Ac2, Ac3, Ac5, Ac6, Ac7, Ac8, Ac9), 4 terpenes (T1, T3, T4, T5), 2 furanones (F2, F3), 3 lactones (L1, L2, L3) 1 other (O1) and 3 phenolic compounds (P2, P3 and P8) (Figure 3B). 

In the PCA score plot, Red-H3 strawberries presented negative PC1 and positive PC2, and were placed in the high left section of the PCA plot (Figure 3A). This sample showed a significant correlation with 2 phenolic compounds (P1 and P4) and 18 VOCs, including 12 esters (E1, E2, E3, E4, E5, E6, E7, E8, E11, E12, E14, E22), 1 aldehydes (Ald1), 2 alcohols (Al3 and Al5), 2-methylbutyric acid (Ac4), β-farnesene (T2) and mesifurane (F1) (Figure 3B). 

The correlation of anthocyanins (P2-P4) with red samples reflects the evident accumulation of these metabolites during ripening, which has been determined for the strawberry on a biosynthetic basis [35]. Interestingly, the increased production of P1 (i.e., *p*-coumaryl-glucoside, a conjugated monolignol) parallels the decline in flavonoids, since the *p*-coumaroyl-CoA precursor is common to the metabolic routes leading to either flavonoid or monolignol biosynthetic pathways. Thus, the downregulation of the flavonoid pathway corresponds to increased levels of monolignols deriving from the conversion of *p*-coumaroyl-CoA [36].

PCA clearly highlighted that only the red samples (Red-H1, Red-H2 and Red-H3) were directly correlated to volatile components (Figure 3B), in agreement with previous reports demonstrating a considerable increase both in number and content of VOCs during the ripening of strawberries [3,30,31,32]. In particular, Red-H1 and Red-H2 samples were statistically associated with a higher number of volatile metabolites (39) compared to Red-H3 strawberries (18) (Figure 3B).

Ester compounds were the most affected by fruit ripening, and they actually characterize mature berries as they are responsible for fruity aromas. Nevertheless, only some of them may be useful as indicators of full ripeness [3]. Among the 14 esters positively associated with the Red-H1 and Red-H2 samples, *trans*-2-hexen-1-ol acetate (E16) and *trans*-2-hexenyl butyrate (E20) were the most abundant esters at the red stage (Table 3), while methyl hexanoate (E9) has been considered as a volatile marker related to the degree of maturity in different strawberry cultivars. Moreover, since it has been reported to be show its highest content in the full red strawberry, E9 has been closely associated with the flavor of mature strawberries [3]. On the other hand, methyl octanoate (E17) has been tentatively identified in the “Candonga” strawberries as responsible for the tropical/pineapple/citrus/green odor [29]. Finally, hexyl acetate (E13) has been previously described among the main contributors to the overall appreciation of strawberry quality [37].

Red-H1 and Red-H2 samples were statistically correlated to the aldehydes Ald2, Ald3, Ald4, Ald5 (Table 3; Figure 3B). Specifically, nonanal (Ald3) and benzaldehyde (Ald4) have been recently identified as volatiles, which enhance sweetness independently of the sugar content together with γ-decalactone (L2) and γ-dodecalactone (L3), which are among the three lactones directly correlated to the Red-H1 and Red-H2 samples (Figure 3B) [38,39]. Specifically, γ-decalactone (L2) gives the major contribution to the desirable “fruity”, “sweet” or “peach-like” aroma in strawberries [40]. In general, several lactone compounds are commonly described as key odorants, determining the pleasant and fruity notes to strawberries [28].

Considering the nine carboxylic acids (Ac1, Ac2, Ac3, Ac5, Ac6, Ac7, Ac8 and Ac9) positively associated to the Red-H1 and Red-H2 samples, hexanoic acid (Ac5) was among the highest abundant odorants in strawberries at the red stage (Table 3). Ac5 has been identified as a key aroma compound of strawberry fruit [37].

Concerning the terpenes group, linalool (T1) also increased with ripeness (Table 3). To this purpose, previous studies pointed at this terpene as an indicator to measure the ripeness stage of fresh strawberry fruits, offering a valuable indication of ideal harvest timing. Nevertheless, it is important to underline that these findings might not be transferable to other cultivars [5].

Finally, among the volatile metabolites positively related to the Red-H1 and Red-H2 strawberries, there was furaneol (F2) already reported in several strawberry varieties, including “Candonga” [29]. F2 is considered a key aroma volatile in strawberry; it increases with ripening, conferring the characteristic caramel-like, sweet, floral, and fruity odor [29,41].

Regarding the 12 esters directly correlated to the Red-H3 strawberries (Figure 3B), methyl butyrate (E2), ethyl butyrate (E4), ethyl pentanoate (E8) and ethyl hexanoate (E11) have been reported as prominent components in several strawberry cultivars and historically recognized as crucial to the characteristic fruity notes of mature strawberries [41,42]. 

Among the volatile metabolites positively associated with the Red-H3 samples, there were also octanol (Al5), previously reported in fruits at late stages of maturation, and mesifurane (F1), which is known to increase during the fruit ripening imparting a particular caramel or cotton candy-like odor in red strawberries [5,37].

Recent studies have established that specific VOCs can enhance both the flavor and the sweetness perception in fresh strawberries, evaluating the overall acceptance of strawberry fruit by the statistical correlations among sensory attributes, such as ‘sweet’ and ‘aromatic’, assessed by a consumers’ panel, and analytical data, including the VOCs profile [38,42]. Regardless of the sugar-acid balance, VOCs with a positive impact on the consumers’ acceptance generally are some esters, including E10, E11 and E13, some aldehydes, such as Ald3, terpenes and lactones, which in our study were positively associated with the Red-H1 and Red-H2 strawberries, while negative coefficients related to liking occur with branched esters, including E5 and E12, and mesifurane (F1), which are directly correlated to the Red-H3 samples (Figure 3B) [38,42]. According to these findings, the Red-H1 and Red-H2 harvesting times could be preferred by the consumers compared to the Red-H3 fruit. 

## 4. Conclusions

In this study, the changes in fruit quality traits, content of VOCs and phenolic compounds in strawberries (cv Sabrosa), commercially referred to as “Candonga”, were evaluated at three consecutive harvesting times. Fruits collected at different times did not present significant differences among the determined qualitative traits, which, on the other hand, significantly differentiated the two ripening stages. Multivariate analysis carried out considering all the chemical data highlighted that Red-H1 and Red-H2 samples were similar to each other, while Red-H3 significantly differed, probably indicating overripe or an early stage of fruit involution. Some volatiles with a positive impact on the consumers’ preference, including E10, E11, E13, Ald3, terpenes and lactones, were positively associated with the Red-H1 and Red-H2 samples, while volatiles with a detrimental contribution to the aroma traits, such as E5, E12 and F1, were directly correlated to the Red-H3 samples. However, although no univocal optimal patterns of volatile metabolites can be associated with a high level of acceptance, and taking into account that the contribution of individual VOCs to a typical aroma relies on both the aroma threshold value (ATV) and its concentration, a correlation between sensory parameters and the VOC patterns could help discover associations within the multiple factors of complex biosystems, such as fruit. For these reasons, the characterization of the VOCs related to the sensory perception of red or half-red samples of “Candonga” strawberries requires the integration with forthcoming experiments based on sensorial perception by a panel test.

In conclusion, the explorative results reported here might be of interest for producers, to identify putative markers useful for the objective assessment of the ripening stage, to optimize the strawberry harvest with the final aim to meet consumers’ liking. 

## Figures and Tables

**Figure 1 foods-10-03102-f001:**
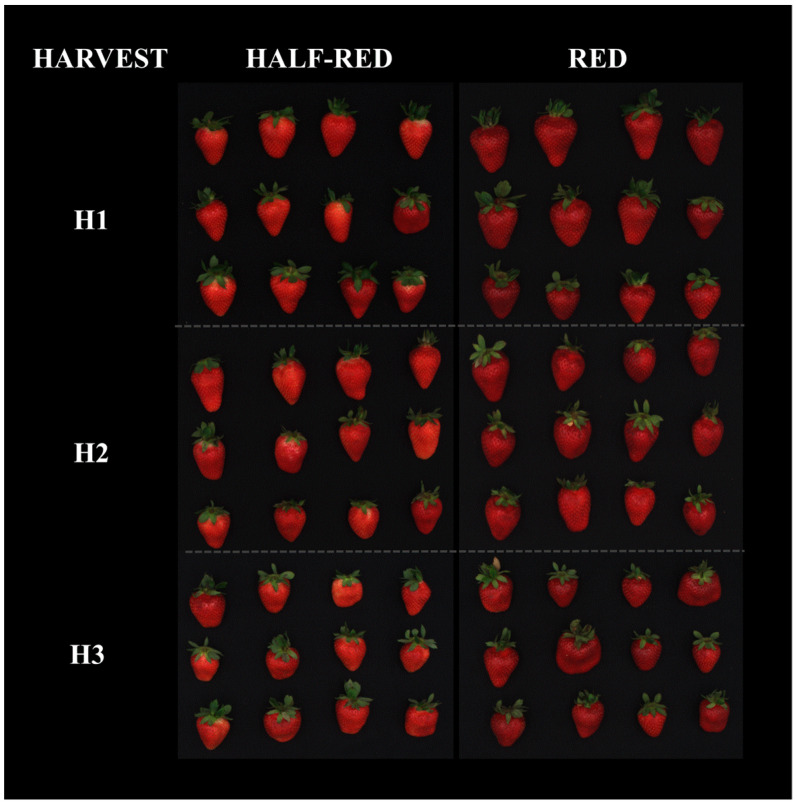
“Candonga” strawberries (var. Sabrosa) harvested at two different ripening stages, namely half-red (in ripening phase, fully expanded and 50% red, indicated as Half-red-H1, Half-red-H2 and Half-red-H3) and red (in ripening phase, fully expanded and 100% red, indicated as Red-H1, Red-H2 and Red-H3).

**Figure 2 foods-10-03102-f002:**
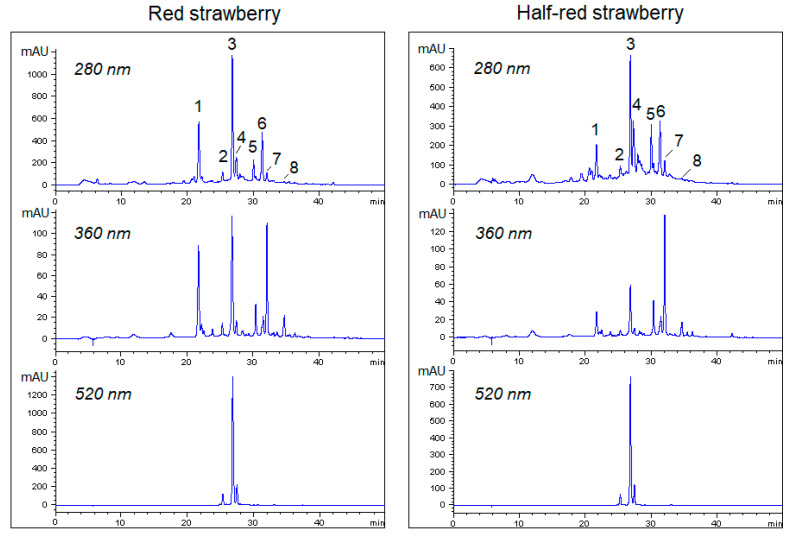
RP-HPLC chromatograms of extracts from red (**left**) and half-red (**right**) “Candonga” strawberries. Numbers (1–8) above the peaks correspond, respectively, to the codes P1–P8 reported on Table 2.

**Figure 3 foods-10-03102-f003:**
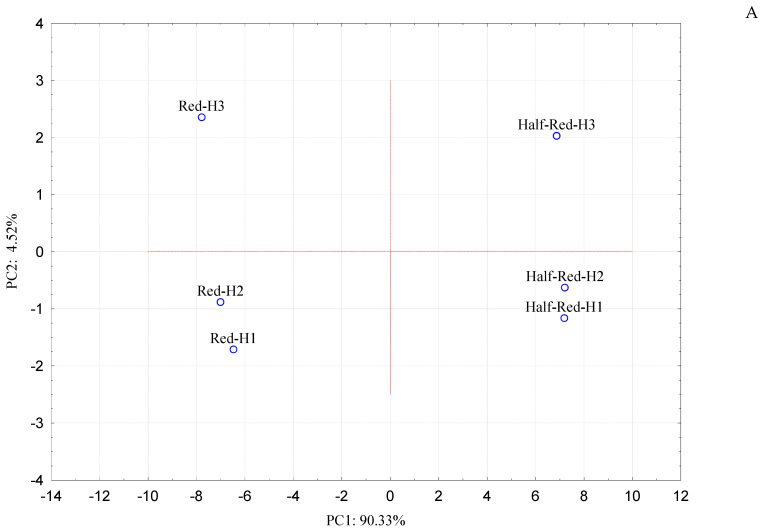
Principal component analysis (PCA) score (**A**) and loading (**B**) plots.

**Table 1 foods-10-03102-t001:** Physical and chemical parameters measured in strawberries cv “Sabrosa” at two different ripening stages (half-red or red) and at three harvest times (H1, H2, H3).

Parameters	Harvest Time														
	H1					H2					H3				
	Ripening Stage														
	Red		Half-Red		*p*	Red		Half-Red		*p*	Red		Half-Red		*p*
Respiration rate (mL CO_2_ kg^−1^ h^−1^)	16.74		14.70		ns	15.33		13.61		ns	16.50		14.00		ns
Titratable acidity (% citric acid)	0.78	b	0.99	a	****	0.86	b	1.00	a	****	0.76	b	0.97	a	****
pH	3.56	a	3.31	b	****	3.44	a	3.27	b	***	3.54	a	3.46	b	*
Total soluble solids (°Brix)	9.61	a	8.37	b	***	9.83	a	8.50	b	****	10.17	a	9.73	b	**
Antioxidant activity (mg Trolox 100 g^−1^ fw)	314.72	a	345.06	b	***	308.12	b	339.11	a	*	287.5	b	347.25	a	***
Total phenols (mg GAE 100 g^−1^ fw )	175.91	a	207.29	b	*	191.46	b	218.05	a	*	183.42	b	207.84	a	*

For each parameter the mean values followed by different letters (a, b) are significantly different (*p* ≤ 0.05) according to least significant difference (LSD) test. Significance: ns = not significant; **** significant for *p* ≤ 0.0001; *** significant for *p* ≤ 0.001; ** significant for *p* ≤ 0.01; * significant for *p* ≤ 0.05. GAE is Gallic Acid Equivalent.

**Table 2 foods-10-03102-t002:** Assignment and semi-quantitative determination of phenolic compounds measured in strawberries cv “Sabrosa” at two different ripening stages (half-red or red) and at three harvest times (H1, H2, H3).

Polyphenols	Code	H1					H2					H3				
		Half-Red		Red		*p*	Half-Red		Red		*p*	Half-Red		Red		*p*
*p*-coumaryl hexoside	P1	9.6	b	28.5	a	****	11.4	b	25.9	a	****	12.7	b	29.0	a	****
cyanidin-3-*O*-glucoside	P2	1.8	b	3.3	a	****	2.0	b	3.8	a	***	1.5	b	3.1	a	***
pelargonidin 3-*O*-glucoside	P3	25.1	b	48.7	a	****	19.8	b	42.6	a	****	22.4	b	43.6	a	****
pelargonidin 3-*O*-rutinoside	P4	3.1	b	5.6	a	****	2.4	b	4.5	a	***	3.5	b	5.1	a	***
quercetin-3-*O*-glucoside	P5	5.1	a	4.2	b	****	4.1	a	3.6	b	*	3.3	a	3.2	b	*
kaempferol-3-*O*-glucoside	P6	2.1	b	2.3	a	*	2.5	a	2.1	b	***	2.1	b	2.3	a	*
quercetin-3-*O*-glucuronide	P7	20.2	a	17.0	b	***	18.6	a	15.8	b	****	12.5	b	14.1	a	***
kaempferol-3-*O*-glucuronide	P8	2.9	b	3.1	a	**	3.0	b	3.4	a	**	2.5	b	2.8	a	**

For each parameter the mean values followed by different letters (a, b) are significantly different (*p* ≤ 0.05) according to least significant difference (LSD) test. Significance: ns = not significant; **** significant for *p* ≤ 0.0001; *** significant for *p* ≤ 0.001; ** significant for *p* ≤ 0.01; * significant for *p* ≤ 0.05.

**Table 3 foods-10-03102-t003:** Volatile compounds obtained in strawberries cv “Sabrosa” at two different ripening stages (half-red or red) and at three harvest times (H1, H2, H3).

Volatile Componds	Ripening Stage															
		H1				H2					H3					
	Code	Red		Half-Red		*p*	Red		Half-Red		*p*	Red		Half-Red		*p*
Methyl propionate	E1	1.92	a	0.91	b	*	2.65	a	0.94	b	****	2.57	a	0.90	b	****
Methyl butyrate	E2	682.95	a	246.38	b	***	806.88	a	256.15	b	****	812.99	a	266.61	b	****
Methyl isovalerate	E3	6.65	a	4.45	b	**	7.70	a	3.97	b	****	7.89	a	4.04	b	****
Ethyl butyrate	E4	46.52	a	4.77	b	*	63.08	a	4.24	b	****	61.06	a	4.29	b	****
Isopropyl butyrate	E5	58.07	a	3.88	b	****	73.39	a	3.62	b	****	75.59	a	4.08	b	****
Butyl acetate	E6	4.18	a	0.00	b	**	4.77	a	0.00	b	***	5.99	a	0.00	b	***
Methyl pentanoate	E7	18.08	a	0.00	b	****	28.53	a	0.00	b	****	28.44	a	0.00	b	****
Ethyl pentanoate	E8	1.75	a	0.00	b	**	4.51	a	0.00	b	****	4.63	a	0.00	b	****
Methyl hexanoate	E9	472.43	a	188.74	b	**	444.88	a	181.44	b	****	465.00	a	182.94	b	****
Butyl butyrate	E10	48.35	a	0.00	b	****	48.95	a	0.00	b	****	49.57	a	0.00	b	****
Ethyl hexanoate	E11	219.48	a	0.00	b	****	227.68	a	0.00	b	****	278.43	a	0.00	b	****
Isoamyl butyrate	E12	3.84	a	0.00	b	**	5.76	a	0.00	b	****	5.07	a	0.00	b	****
Hexyl acetate	E13	473.52	a	35.15	b	****	446.02	a	35.86	b	****	441.31	a	35.77	b	****
Methyl 2-hexenoate	E14	21.51	a	3.86	b	****	21.07	a	3.48	b	****	24.98	a	11.61	b	****
*cis*-3-Hexen-1-ol acetate	E15	39.92	a	4.63	b	****	39.40	a	3.97	b	****	39.57	a	3.82	b	****
*trans*-2-Hexen-1-ol acetate	E16	2724.77	a	142.74	b	****	2664.89	a	141.33	b	****	2688.80	a	141.61	b	****
Methyl octanoate	E17	42.43	a	4.64	b	****	41.15	a	4.73	b	****	41.56	a	4.72	b	****
*trans*-2-Hexen-1-ol propionate	E18	93.04	a	4.49	b	****	91.79	a	4.51	b	****	91.15	a	4.72	b	****
n-Hexyl isobutyrate	E19	520.24	a	6.75	b	****	515.75	a	6.89	b	****	511.47	a	6.76	b	****
*trans*-2-Hexenyl butyrate	E20	1114.73	a	21.94	b	****	1107.06	a	21.32	b	****	1080.35	a	21.21	b	****
Methyl 3-(methylthio) propionate	E21	93.01	a	2.19	b	****	91.28	a	2.33	b	****	92.29	a	2.18	b	****
Hexyl hexanoate	E22	4.29	a	1.18	b	****	4.10	a	1.29	b	****	4.48	a	1.40	b	****
n-Octyl isobutyrate	E23	11.63	a	0.00	b	****	11.31	a	0.00	b	****	11.42	a	0.00	b	****
Octyl 2-methylbutyrate	E24	4.03	a	0.00	b	****	4.00	a	0.00	b	****	4.00	a	0.00	b	****
Methyl 3-hydroxyhexanoate	E25	4.62	a	1.67	b	****	4.64	a	1.74	b	****	4.07	a	1.73	b	****
Benzyl acetate	E26	15.53	a	7.49	b	****	15.66	a	7.66	b	****	15.93	a	7.65	b	****
Hexanal	Ald1	19.41	a	11.47	b	*	33.11	a	11.96	b	****	33.88	a	12.69	b	****
2-Hexenal	Ald2	532.42	a	234.98	b	***	503.36	a	221.54	b	****	523.03	a	219.89	b	****
Nonanal	Ald3	29.57	a	2.22	b	****	26.25	a	2.46	b	****	26.75	a	2.64	b	****
Benzaldehyde	Ald4	76.65	a	5.12	b	****	77.97	a	5.22	b	****	77.14	a	5.37	b	****
Dodecanal	Ald5	8.24	a	3.42	b	****	8.08	a	3.66	b	****	8.14	a	3.60	b	****
1-Hexanol	Al1	209.84	a	27.21	b	****	209.38	a	28.10	b	****	215.20	a	27.69	b	****
*trans*-3-Hexen-1-ol	Al2	11.12	a	2.02	b	****	11.08	a	2.11	b	****	11.89	a	2.18	b	****
*cis*-3-Hexen-1-ol	Al3	11.69	a	4.92	b	****	12.38	a	4.96	b	****	12.79	a	4.95	b	****
*trans*-2-Hexen-1-ol	Al4	419.31	a	110.29	b	****	404.16	a	111.24	b	****	402.86	a	110.18	b	****
1-Octanol	Al5	3.47	a	1.31	b	****	3.03	a	1.31	b	****	3.32	a	1.69	b	****
Propanoic acid	Ac1	37.29	a	1.56	b	****	37.87	a	1.69	b	****	37.72	a	1.78	b	****
2-Methylpropionic acid	Ac2	70.11	a	1.30	b	****	70.47	a	1.30	b	****	71.67	a	1.33	b	****
Butyric acid	Ac3	24.85	a	8.13	b	****	24.15	a	8.20	b	****	24.27	a	8.31	b	****
2-Methylbutanoic acid	Ac4	80.09	a	12.99	b	****	88.70	a	12.82	b	****	733.19	a	12.06	b	****
Hexanoic acid	Ac5	1423.08	a	332.86	b	****	1424.30	a	331.78	b	****	1426.72	a	332.20	b	****
Heptanoic acid	Ac6	33.53	a	4.26	b	****	33.57	a	4.22	b	****	33.89	a	4.32	b	****
Octanoic acid	Ac7	35.88	a	5.99	b	****	35.50	a	6.05	b	****	35.56	a	6.19	b	****
Nonanoic acid	Ac8	56.54	a	19.18	b	****	56.05	a	19.35	b	****	56.06	a	19.15	b	****
Decanoic acid	Ac9	28.59	a	2.52	b	****	29.25	a	2.50	b	****	29.71	a	2.54	b	****
Linalool	T1	111.83	a	62.76	b	****	112.66	a	62.20	b	****	112.18	a	62.92	b	****
β-Farnesene	T2	11.12	a	0.00	b	****	8.88	a	0.00	b	****	11.75	a	0.00	b	****
α-Terpineol	T3	98.36	a	10.26	b	****	98.21	a	10.86	b	****	98.37	a	10.57	b	****
β-Damascenone	T4	1.47	a	0.92	b	****	1.47	a	0.92	b	****	1.46	a	0.91	b	****
Nerolidol	T5	130.56	a	6.56	b	****	130.15	a	6.21	b	****	130.22	a	6.29	b	****
Mesifurane	F1	140.90	a	12.89	b	****	133.47	a	12.81	b	****	1837.09	a	12.65	b	****
Furaneol	F2	42.61	a	3.28	b	****	41.99	a	3.26	b	****	42.37	a	3.19	b	****
*trans*-γ-Jasmolactone	F3	46.65	a	0.00	b	****	46.69	a	0.00	b	****	46.48	a	0.00	b	****
γ-Octalactone	L1	4.31	a	0.00	b	****	4.44	a	0.00	b	****	4.57	a	0.00	b	****
γ-Decalactone	L2	2689.87	a	262.49	b	****	2631.41	a	260.20	b	****	2660.37	a	260.14	b	****
γ-Dodecalactone	L3	60.16	a	11.62	b	****	61.88	a	11.00	b	****	59.96	a	11.64	b	****
Acetophenone	O1	4.42	a	0.00	b	****	4.47	a	0.00	b	****	4.52	a	0.00	b	****

For each parameter the mean values followed by different letters (a, b) are significantly different (*p* ≤ 0.05) according to least significant difference (LSD) test. Significance: ns = not significant; **** significant for *p* ≤ 0.0001; *** significant for *p* ≤ 0.001; ** significant for *p* ≤ 0.01; * significant for *p* ≤ 0.05.

## Data Availability

The datasets generated for this study are available on request to the corresponding author.

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
