# Peer review of "Profiles of Volatile and Phenolic Compounds as Markers of Ripening Stage in Candonga Strawberries"

_foods, 2021, doi:10.3390/foods10123102_

Round 1

Reviewer 1 Report

Could you please write before spectrophotometric and HPLC analysis a separate section of sample preparation (how you prepare methanolic extracts) with the title for E. g. Preparation of methanol extract 

Please write in italic labels for oxygen (cyanidin-3-O-glucoside), para-, trans- etc.

Author Response

Reviewer 1.

Point 1. Could you please write before spectrophotometric and HPLC analysis a separate section of sample preparation (how you prepare methanolic extracts) with the title for E. g. Preparation of methanol extract 

Response to Point 1.

The preparation of methanolic extracts (80% aqueous methanol, v/v) has been already described with relevant details (Par. 2.5). To avoid redundancies, we did not include a novel section. However, to meet the reviewer’s suggestion, in the paragraph 2.6 (Analysis of polyphenols compounds), precisely in the section 2.6.1, we have specified that extracts were prepared as detailed in Par. 2.5.

Point 2. Please write in italic labels for oxygen (cyanidin-3-O-glucoside), para-, trans- etc.

Response to Point 2.These points have been corrected, accordingly.

Reviewer 2 Report

The manuscript by Cozzolino and colleagues reports about the role of Volatile and phenolic compounds as markers of ripening stage in Candonga strawberries. The authors demonstrate that volatiles with positive impact on the consumers’ acceptance clearly increased both in number and amount during ripening. Furthermore, they provide comprehensive pattern of polyphenols in strawberry fruit at two different ripening stages and at three harvest times. However, after thoroughly reading the manuscript there are concerns that should be addressed to present a coherent story and further improve the manuscript.

  1. The authors should modify the abstract to include the results of phenolic compounds as putative marker of the ripening stage in Candonga strawberries.
  2. The introduction focuses only on VOCs during the fruit maturation process. The authors should modify the introduction to include more information on the current knowledge available on the acquisition of phenolic compounds with fruit quality.
  3. The results and discussion main focuses on VOCs during the fruit maturation process. The authors should provide additional results to show the phenolic compounds correlated to ripening stage of strawberry, or modify the article tittle.

Author Response

Reviewer 2.

The manuscript by Cozzolino and colleagues reports about the role of Volatile and phenolic compounds as markers of ripening stage in Candonga strawberries. The authors demonstrate that volatiles with positive impact on the consumers’ acceptance clearly increased both in number and amount during ripening. Furthermore, they provide comprehensive pattern of polyphenols in strawberry fruit at two different ripening stages and at three harvest times. However, after thoroughly reading the manuscript there are concerns that should be addressed to present a coherent story and further improve the manuscript.

Point 1. The authors should modify the abstract to include the results of phenolic compounds as putative marker of the ripening stage in Candonga strawberries.

Point 2. The introduction focuses only on VOCs during the fruit maturation process. The authors should modify the introduction to include more information on the current knowledge available on the acquisition of phenolic compounds with fruit quality.

Point3. The results and discussion main focuses on VOCs during the fruit maturation process. The authors should provide additional results to show the phenolic compounds correlated to ripening stage of strawberry, or modify the article title.

Response to Point 1-3.  In the present study, authors did not identify specific polyphenol markers of ripening, intended as “on/off” compounds associated with the ripening stage. In contrast, quali/quantitative variations of the profiles of phenolic compounds, depending on the maturation stage, have been highlighted.

Authors provided information about the evolving balance of HPLC-separated individual phenolic compounds during ripening. The polyphenol compounds involved in the changes have been indicated in the Abstract of the revised version of the manuscript. The ripening-related metabolic switch affecting the composition of polyphenol compounds has been specified in Results and Discussion, quoting several literature articles specifically focused on the activation of metabolic pathways during strawberry maturation (Paragraphs 3.2 and 3.3.2). Finally, title has been modified accordingly.

Reviewer 3 Report

The study is relevant to the Journal’s scope and provides new data which are interesting from an applicative viewpoint as well as scientific soundness, while the manuscript is well written. A series of analyses were well performed, data analyses were performed correctly, and data interpretations were true to the results. 

  • Replace keywords that already exist in the title
  • Lines 57-58, clarify better why non climacteric fruit are harvesting earlier than the optimum maturity stage.
  • Lines 90-91, PET trays manufacturer?
  • Please clarify the need of 3 harvests? Probably to consider as experiment replication. Also, there were significant difference on environmental conditions before each harvest?

Author Response

Reviewer 3.

The study is relevant to the Journal’s scope and provides new data which are interesting from an applicative viewpoint as well as scientific soundness, while the manuscript is well written. A series of analyses were well performed, data analyses were performed correctly, and data interpretations were true to the results. 

Point 1. Replace keywords that already exist in the title

Response to Point 1. Done

Point 2. Lines 57-58, clarify better why non climacteric fruit are harvesting earlier than the optimum maturity stage.

Response to Point 2. Strawberry is classified as non-climacteric fruit. Moreover, its shelf life is limited, generally when the fruits achieve the optimum maturity stage their shelf life reach about 10 days.

Strawberry fruits are particularly appreciate by consumer when they are collected at their optimal maturity stage (full red and over), but at this stage they are more susceptible to pathogens that limit their commerciality during supply chain.

Thus, for commercial reasons, the harvest starts when the fruit colour is half red. In the present research paper strawberries were collected and analysed at two maturity stage as required by the producer farm (Apofruit (Scanzano Jonico, Italy).

Point 3. Lines 90-91, PET trays manufacturer?

Response to Point 3. Done

Point 4. Please clarify the need of 3 harvests? Probably to consider as experiment replication. Also, there were significant difference on environmental conditions before each harvest?

Response to Point 4. In order to have the experiment replicate three times, three harvesting times were considered in a very short period (10 days from the first to the end) for the two different stages of maturation. The harvests were made on the same farm. No changes in environmental conditions were recorded during the harvest period.

Round 2

Reviewer 2 Report

The author claim that the ripening-related metabolic switch affecting the composition of polyphenol compounds has been specified in Results and Discussion. Where? Please highlight.

Author Response

Point 1: The author claim that the ripening-related metabolic switch affecting the composition of polyphenol compounds has been specified in Results and Discussion. Where? Please highlight.

Response to Point 1:

As required by Reviewer, we highlighted in red the part of the manuscript in which the ripening-related metabolic switch affecting the composition of polyphenol compounds has been specified.

These part are also reported below:

Par. 3.2

In agreement with previous data, p-coumaryl-hexoside (peak 1, P1) was the most abundant hydroxycinnamic acid derivative of strawberry, better detected at 320 nm (not shown), because of the characteristic absorbance band centered at 315 nm. In half-red strawberries, P1 was 3-/4-fold lower than in the red samples.

However, the concentration of anthocyanins in unripe fruits was nearly half than the red counterpart. The chromatogram at 360 nm was dominated by quercetin-3-O-glucuronide (P7), which was more abundant in Half-red-H1 and Half-red-H2, than in the ripened counterpart.

Par 3.3.2

The direct association of Half-red-H1 and Half-red-H2 samples with the P5, P6 and P7 metabolites likely reflects the decline of the synthesis of flavonoids during the last stage of ripening, as demonstrated for other fruits. The decrease of flavonoids might correspond to their utilization for the downstream biosynthesis of other metabolites or to the covalent association with other cellular components [34].

The correlation of anthocyanins (P2-P4) with red samples reflects the evident accumulation of these metabolites during ripening, which have been determined for strawberry on a biosynthetic basis [35]. Interestingly, the increased production of P1 (i.e., p-coumaryl-glucoside, a conjugated monolignol) parallels the decline of flavonoids, since the p-coumaroyl-CoA precursor is common to the metabolic routes leading to either flavonoid or monolignol biosynthetic pathways. Thus, the downregulation of the flavonoid pathway corresponds to increased levels of monolignols deriving from the conversion of p-coumaroyl-CoA [36]
